# A comparison study of temporal trends of SARS-CoV2 RNAemia and biomarkers to predict success and failure of high flow oxygen therapy among patients with moderate to severe COVID-19

Hiroshi Koyama[1]*, Kazuya Sakai[2], Kiyomitsu Fukaguchi[1,3], Hiroki Hadano[1], Yoshihisa Aida[3], Tadashi Kamio[1], Takeru Abe[2], Mototsugu Nishii[2], Ichiro Takeuchi[2]

1 Department of Critical Care Medicine, Shonan Kamakura General Hospital, Kamakura, Japan,
2 Department of Emergency Medicine, Yokohama City University School of Medicine, Yokohama, Japan,
3 Center for Acute and General Medicine, Shonan Kamakura General Hospital, Kamakura, Japan

* ko1978hiro@gmail.com

**Data Availability Statement:** The study did not receive the ethical approval under the assumption

## Abstract

Optimal timing for intubating patients with coronavirus disease 2019 (COVID-19) has been debated throughout the pandemic. Early use of high-flow nasal cannula (HFNC) can help reduce the need for intubation, but delay can result in poorer outcomes. This study examines trends in laboratory parameters and serum severe acute respiratory syndrome coronavirus 2 (SARS-CoV-2) RNA levels of patients with COVID-19 in relation to HFNC failure. Patients requiring HFNC within three days of hospitalization between July 1 and September 30, 2021 were enrolled. The primary outcome was HFNC failure (early failure ≤Day 3; late failure ≥Day 4), defined as transfer to intensive care just before/after intubation or in-hospital death. We examined changes in laboratory markers and SARS-CoV2-RNAemia on Days 1, 4, and 7, together with demographic data, oxygenation status, and therapeutic agents. We conducted a univariate logistic regression with the explanatory variables defined as 10% change rate in each laboratory marker from Day 1 to 4. We utilized the log-rank test to assess the differences in HFNC failure rates, stratified based on the presence of SARS-CoV2 RNAemia. Among 122 patients, 17 (13.9%) experienced HFNC failure (early: n = 6, late: n = 11). Seventy-five patients (61.5%) showed an initial $SpO_2/FiO_2$ ratio ≤243, equivalent to $PaO_2/FiO_2$ ratio ≤200, and the initial $SpO_2/FiO_2$ ratio was significantly lower in the failure group (184 vs. 218, p = 0.018). Among the laboratory markers, a 10% increase from Day 1 to 4 of lactate dehydrogenase (LDH) and interleukin (IL)-6 was associated with late failure (Odds ratio [OR]: 1.42, 95% confidence interval [CI]: 1.09–1.89 and OR: 1.04, 95% CI: 1.00–1.19, respectively). Furthermore, in patients with persistent RNAemia on Day 4 or 7, the risk of late HFNC failure was significantly higher (Log-rank test, p<0.01). In conclusion, upward trends in LDH and IL-6 levels and the persistent RNAemia even after treatment were associated with HFNC failure.

of being used as an open-source database. The data contains individual-level information, and the informed consent sheet and participant information disclosure document did not include information about registering data for public availability. The data are available from the Department of Critical Care Medicine (contact via the corresponding author: ko1978hiro@gmail.com) or the Center for Clinical Research (contact: ccts512@shonankamakura.or.jp) at the Shonan Kamakura General Hospital for researchers who meet the criteria for access to confidential data. The data are stored and preserved in the independent hard disc drives in the facility.

**Funding:** Funding was obtained from Japan Agency for Medical Research and Development (AMED 20fk0108405h0001). The funders had no role in study design, data collection and analysis, decision to publish, or preparation of the manuscript.

**Competing interests:** The authors have declared that no competing interests exist.

## Introduction

The optimal timing to intubate patients with coronavirus disease 2019 (COVID-19) exhibiting acute hypoxemia has been a subject of debate throughout the pandemic [1–9]. In the early stages of the pandemic, it was recommended to intubate patients with COVID-19 very early when their oxygen requirements exceeded 6L/min, primarily as a measure to prevent the spread of the virus [1, 2]. However, subsequent evidence regarding the potential benefits of high-flow nasal cannula (HFNC) oxygen therapy and non-invasive mechanical ventilation, coupled with concerns about the potential harm caused by premature unnecessary intubation, have raised questions regarding the very early intubation strategy [3, 5]. Meanwhile, a meta-analysis including 20 non-randomized controlled studies revealed higher in-hospital mortality in patients that were intubated 24–48 hours or later after admission [8]. Nevertheless, it is important to note that all the studies included in this analysis were retrospective observational studies, and the precise criteria for intubation remain ambiguously defined. Furthermore, the meta-analysis did not consider the responsiveness to initial pharmacological treatment. Another study investigated intubated patients and divided them into groups based on when patients received dexamethasone treatment [9]. They found that patients intubated after seven days of dexamethasone treatment had lower lung function and a higher risk of in-hospital mortality compared to those intubated before, on Day 1, or between Days 1 and 7 of dexamethasone treatment. The last two groups had similar mortality rates. This study emphasized that when considering the optimal timing for intubation, it is crucial to factor in the progression of the disease and the response to treatment, rather than solely relying on the passage of time. Although striking the right balance between not intubating too early or too late is essential, no definitive objective parameters exist at present.

Early initiation of HFNC can contribute in preventing the need for intubation and invasive mechanical ventilation (IMV), with additional effects of reducing the work of breathing and providing a stable supply of high-concentration oxygen [10–12]. However, the specific feature of "happy hypoxia," a phenomenon in which the patient feels less respiratory discomfort despite hypoxemia [13], has confused clinicians regarding the appropriate timing of intubation under HFNC treatment. Several studies evaluated the respiratory rate-oxygen (ROX) index ([$SpO_2/FiO_2$]/respiratory rate) in the early hours after HFNC induction for predicting HFNC failure. Nevertheless, the intubation threshold and the cut-off level of the ROX index varied between the studies [14]. We, therefore, hypothesized that, in addition to the physiological parameters, chronological changes in laboratory parameters and severe acute respiratory syndrome coronavirus 2 (SARS-CoV-2) viral load before and after pharmacological treatment intervention, would be more accurate in determining the optimal timing of intubation under HFNC treatment. The severity and progression of the disease have been documented through various laboratory parameters, including the neutrophil/lymphocyte (NL) ratio, C-reactive protein (CRP), interleukin-6 (IL-6), lactate dehydrogenase (LDH), ferritin and D-dimer, as well as the presence of SARS-CoV2 RNAemia, which indicates the presence of SARS-CoV-2 viral RNA in the bloodstream [15–18]. Nonetheless, no studies have compared longitudinal trends of these laboratory parameters and SARS-CoV-2 RNAemia in relation to the success or failure of HFNC therapy as the disease progresses.

To clarify the optimal timing for escalating respiratory support, we retrospectively investigated laboratory parameters trends and SARS-CoV2 RNAemia among patients with moderate to severe COVID-19. In this context, we implemented the "avoid intubation strategy"—a structured HFNC treatment protocol with early induction and extended application, at a temporary field hospital without an intensive care unit (ICU) during the pandemic wave dominated by the delta-variant, when the regional ICU capacity was overwhelmed.

## Methods

### Design and setting

This was a retrospective cohort study conducted in a 180-bed prefectural temporary field hospital run by Shonan Kamakura General Hospital, located in Kanagawa prefecture, Japan. The study was a sub-analysis of a Japanese multicenter clinical study investigating comprehensively the prognostic factors of COVID-19. The hospital was exclusively designated to care for adult patients with moderate to severe COVID-19, except for pregnant or mechanically ventilated patients [19]. The hospital did not have a designated intensive care unit (ICU). However, each bedside was equipped with an oxygen supplementation system via liquid oxygen tanks, a stable power supply, and a vital sign monitor. The patient-to-nurse staffing ratio was approximately 10:1 to 7:1, depending on the number of patients and the severity. Each shift was assigned at least one doctor with expertise in intubation and induction of mechanical ventilation. Under the prefectural regulation, continuous management of mechanically ventilated patients was not permitted due to insufficient ICU-specific equipment and labor constraints.

### Participants

The study included consecutive adult patients ($\geq$18 years old) admitted to the hospital during Japan's fifth pandemic wave from July 1, 2021 until September 30, 2021, and those who commenced HFNC within the first 3 days after admission. The SARS-CoV2 infection was confirmed by either a positive nucleotide acid amplification test or a positive antigen test prior to admission. Patients who required or were deemed likely to require oxygen support were transported to the hospital via ambulance from home, other hospitals, or the emergency department in the Shonan Kamakura General Hospital. Patients requiring immediate resuscitation (e.g., shock or bag-mask ventilation in the prehospital setting) were not transferred to the facility. To only include patients in an initial deteriorating phase, we excluded patients that were transferred from an intensive care facility for step-down treatment.

### Ethical considerations

This study was conducted according to the principles expressed in the Declaration of Helsinki. The written informed consent was required, and the opt-out approach to consent was also permitted under the public health emergency. The study was approved by the Tokushukai Group Ethics Committee (TGE01640-024). Patient information was anonymized and entered into a multicenter database. The anonymized dataset was finally assessed on August 31, 2023. Access to personally identifiable information for analysis and validation after data collection was restricted to authorized personnel.

### High-flow nasal cannula oxygen therapy in the facility

We established a protocol for HFNC therapy. In summary, we initiated HFNC when the patient's $SpO_2$ could not be maintained at 92% with an oxygen flow rate of approximately 4–6 L/min using a conventional oxygen therapy. All HFNC equipment used in the hospital was AIRVO2® (Fisher & Paykel, New Zealand). The HFNC flow rate was adjusted between 30–60 L/min, and the $FiO_2$ was adjusted between 0.35–0.95 to maintain a $SpO_2$ level of at least 90–92%. The use of HFNC was discontinued when the patient's $SpO_2$ reached 92% or above, with a HFNC flow rate below 30 L/min and a $FiO_2$ below 0.35–0.4. Awake-prone positioning was encouraged for all patients receiving HFNC therapy. If the $FiO_2$ level exceeded 0.6 or above, the clinical team was alerted, and the patient was closely monitored. The final decision regarding intubation or transfer for induction of mechanical ventilation was made by the clinical

team, including experienced intensivists. The decision was reached when $SpO_2$ levels dropped below 90%, with a $FiO_2$ of 0.9 or higher, and concurrent deterioration in excessive work of breathing. Patients who declared a do-not-intubate status continued to receive treatment in the facility.

## Exposures and covariates

Baseline patient information included age, sex, body mass index, smoking status, presence of underlying diseases (hypertension, diabetes, chronic obstructive pulmonary disease, asthma, chronic heart failure, stroke, liver cirrhosis, chronic kidney disease, immunocompromised status, and malignancy), SARS-CoV2 vaccination status, days from onset to admission, and initial vital signs. The initial respiratory condition was presented with $SpO_2/FiO_2$ ratio and ROX index. If conventional oxygen therapy was used, $FiO_2$ was calculated using the following formula: $0.21 + (\text{oxygen flow rate} \times 0.04)$. Estimated $PaO_2/FiO_2$ ratio was calculated from the results of $SpO_2/FiO_2$ ratio according to a previous study [20]: a $PaO_2/FiO_2$ of 200 mmHg corresponded to a $SpO_2/FiO_2$ of 243. All patients underwent a routine chest CT scan upon admission, and diagnosis of pneumonia was made by an experienced doctor specializing in COVID-19 treatment at the facility. We extracted data on the regimen of therapeutic agents administered within the first three days following hospital admission. Although the facility's pharmacological treatment protocol was mostly based on the national treatment guideline, we used a higher dose of dexamethasone (e.g., 20 mg/day) drawing reference from a previous study conducted at that time [21].

Laboratory tests were routinely conducted on Days 1, 4, and 7 following admission. The tests included NL ratio, CRP, IL-6, LDH, ferritin, and D-dimer. When available, serum SARS-CoV-2 RNA was quantified at the same time point using a specimen obtained as part of routine clinical work. The RNA was extracted using the QIAamp Viral RNA Mini Kit (QIAGEN, 52906; Hilden, Germany) following the manufacturer's instructions. Viral genes were quantified by real-time qPCR using N2 primer pairs (TaKaRa, XD0008, forward primer: `AAATTTT GGGGACCAGGAAC`, reverse primer: `TGGCAGCTGTGTAGTCAAC`, probe: `FAM-ATGTCGC GCATTGGCATGGA-BHQ`). The Ct values were plotted on a standard curve using the provided standard (Nihon Gene Research Laboratories, Inc., JP-NN2-PC; Miyagi-ken, Japan). This standard curve was then used to calculate the viral RNA content.

## Primary outcome

The primary outcome was HFNC failure, defined as either transfer of patients to an intensive care facility after intubation or under urgent need of intubation, or in-hospital death by respiratory failure owing to do-not-intubate code status. The HFNC failure was further categorized into two groups based on the timing of the failure: patients who experienced HFNC failure within the first three days of hospitalization (early failure group) and those who experienced failure on or after the fourth day of hospitalization (late failure group).

## Statistical analysis

Continuous variables are presented as medians with an interquartile range and analyzed using the Mann–Whitney $U$ test. Categorical variables were presented as numbers and percentages and analyzed using the Fisher's exact test. We initially compared the overall HFNC success and failure based on the explanatory variables on admission. A serum SARS-CoV2 RNA viral load was log-transformed to improve the normality of the distribution and reduce the influence of outliers.

After excluding the patients who deteriorated within the first 3 days after hospitalization (early failure group), we analyze the association between each laboratory marker trend from Day 1 to Day 4 and the outcome of late HFNC failure. To investigate this association, we conducted a univariable logistic regression with the explanatory variables defined as the 10% change rate in each laboratory marker from Day 1 to Day 4. The calculated odds ratio (OR) was presented with 95% confidence intervals (CI). We plotted the Area Under the Receiver Operating Characteristic (AUROC) curves to predict overall failure based on Day 1 values and late failure based on Day 4 values for each laboratory marker.

Finally, we defined persistent SARS-CoV2 RNAemia as detection of SARS-CoV2 RNA in patient's blood on either Day 4 or Day 7 or both. We performed survival analysis by stratifying patients based on the presence or absence of initial and persistent SARS-CoV-2 RNAemia and evaluated the significance of the difference in survival curves using the log-rank test. Pearson's correlation coefficient was computed for each laboratory parameter in relation to the log-transformed SARS-CoV-2 RNA level on Days 1, 4, and 7 and was presented as "*r*." A *p*-value of less than 0.05 was considered statistically significant. All statistical analysis was performed using R statistical software (R version 4.3.1).

## Results

Overall, 504 patients were hospitalized during the study period. Of these, 122 patients required HFNC within the first 3 days after admission. Six patients (4.9%) experienced HFNC failure within 3 days of admission (early failure), 11 patients (9.0%) experienced HFNC failure 4 days after admission (late failure), and 105 patients (86.1%) successfully discontinued HFNC. The details of the HFNC failure are shown in Fig 1.

Table 1 demonstrates the baseline characteristics between the patients' groups of HFNC success and overall HFNC failure. No statistical difference was found between the HFNC success and overall HFNC failure groups in terms of age (52.0 vs 55.0 years, respectively, *p* = 0.311), sex (female/male; 23.8%/76.2% vs 35.3%/64.7%, respectively, *p* = 0.478), body mass index (26.8 vs 28.1, respectively, *p* = 0.74), smoking habits (current smoker; 17.1% vs 29.4%, respectively, *p* = 0.387), vaccination status (not-vaccinated; 79.0% vs 76.5%, respectively, *p* = 0.734), days from onset to admission (8.0 days vs 7.0 days, respectively, *p* = 0.126), and underlying diseases except for stroke (1.9% vs 17.6%, respectively, *p* = 0.017). All patients revealed lung infiltrates on their initial CT scan. The type of therapeutic agents administered within the first 3 days after admission did not differ between the success and failure groups: remdesivir (94.3% vs 100.0%, respectively, *p* = 0.685), dexamethasone (100.0% vs 100.0%, respectively), baricitinib (72.4% vs 64.7%, respectively, *p* = 0.719), and casirivimab/imdevimab (5.7% vs 0.0%, respectively, *p* = 0.685). Tocilizumab was not used during the study period. A detailed comparison of the three groups (success, early failure, and late failure) is shown in the S1 Table.

An initial $SpO_2/FiO_2$ ratio was higher in the success group compared to the failure group (218 vs 184, respectively, *p* = 0.018). In the study cohort, 61.5% (n = 75) of the patients showed $SpO_2/FiO_2$ ratio of <243, which is equivalent to $PaO_2/FiO_2$ ratio of 200 mmHg. Among these 75 patients, 61 patients (81.3%) successfully discontinued HFNC. An initial ROX index revealed higher tendency in the success group compared to the failure group (9.9 and 8.8, respectively, *p* = 0.071). HFNC was introduced on either Day 1 of admission (n = 75, 61.5%) or Day 2 (n = 30, 24.6%) or Day 3 (n = 17, 13.9%) following the facility's protocol. The median duration of HFNC treatment was 6 days [range: 1–33 days] in the success group, 2 days [range: 1–2 days] in the early failure group, and 11 days [range: 3–50 days] in the late failure group.

Fig 2 and S2 Table demonstrate the trend of each laboratory parameter on Days 1, 4, and 7. Upon admission, no significant differences were found between the overall success and failure

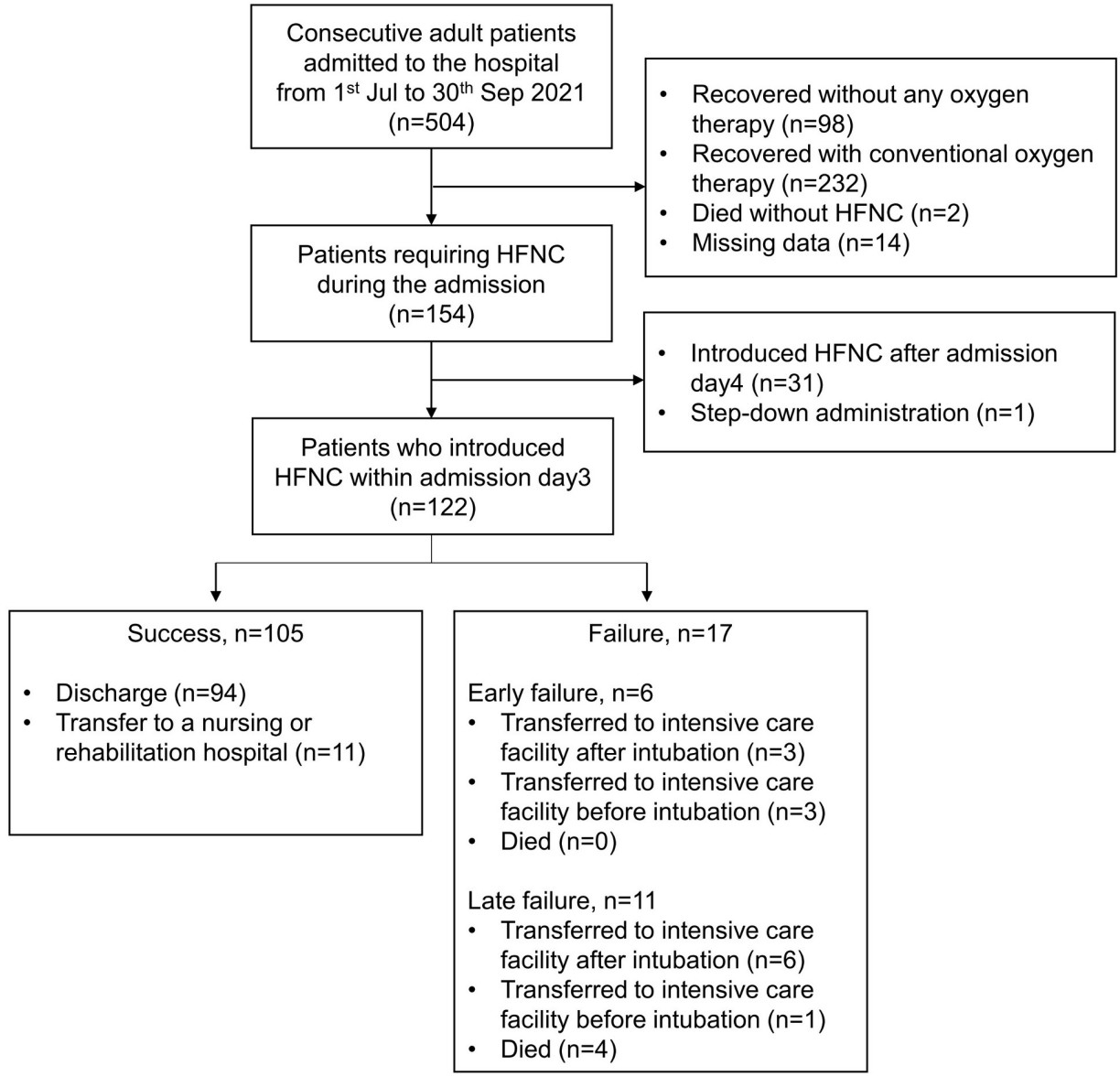

**Fig 1. Study flow chart.**

groups for each laboratory parameter: NL ratio (7.61 vs 11.24, respectively, $p = 0.093$), CRP (11.68 vs 10.29, respectively, $p = 0.408$), IL-6 (69.45 vs 123.0, respectively, $p = 0.102$), LDH (549.0 vs 627.0, respectively, $p = 0.082$), and ferritin (982.30 vs 1254.55, respectively, $p = 0.234$), except for D-dimer (0.80 vs 1.40, respectively, $p = 0.049$). After excluding the early failure group, a significant difference was observed on Day 4 for LDH (465.0 vs. 623.5, respectively, $p = 0.001$) and IL-6 (8.65 vs 23.7, respectively, $p = 0.001$) between the success and late failure groups (S2 Table). In contrast, no significant difference was found in the value of NL ratio (8.82 vs 11.42, respectively, $p = 0.085$), CRP (1.95 vs 1.45, respectively, $p = 0.171$), ferritin (1130.8 vs 1771.2, respectively, $p = 0.347$), and D-dimer (1.10 vs 1.40, respectively, $p = 0.431$) on Day 4. On Day7, following the exclusion of 5 patients who experienced HFNC failure between Day 4 and Day 7, a significant difference was observed in broader laboratory markers:

**Table 1. Baseline characteristics of the patients.**

| | HFNC success (n = 105) | HFNC failure (n = 17) | *p*-value |
|---|---|---|---|
| Age (in years) | 52.0 [46.0–60.0] | 55.0 [50.0–71.0] | 0.311 |
| Sex, Female/Male, n(%) | 25/80 (23.8/76.2) | 6/11 (35.3/64.7) | 0.478 |
| BMI, kg/m$^2$ | 26.8 [24.1–31.6] | 28.1 [23.9–31.7] | 0.74 |
| Current smoker, n(%) | 18 (17.1) | 5 (29.4) | 0.387 |
| Underlying diseases, n(%) | | | |
| Hypertension | 33 (31.4) | 10 (58.8) | 0.055 |
| Diabetes | 46 (43.8) | 10 (58.8) | 0.373 |
| COPD | 4 (3.8) | 0 (0.0) | 0.933 |
| Asthma | 2 (1.9) | 0 (0.0) | 1 |
| CHF | 2 (1.9) | 0 (0.0) | 1 |
| Stroke | 2 (1.9) | 3 (17.6) | 0.017 |
| Liver cirrhosis | 5 (4.8) | 0 (0.0) | 0.795 |
| CKD | 4 (3.8) | 0 (0.0) | 0.933 |
| Immunocompromised | 1 (1.0) | 1 (5.9) | 0.649 |
| Malignancy | 0 (0.0) | 1 (5.9) | 0.296 |
| Number of vaccinations | | | 0.734 |
| 0 | 83 (79.0) | 13 (76.5) | |
| 1 | 18 (17.1) | 3 (17.6) | |
| 2 | 2 (1.9) | 0 (0.0) | |
| not recorded | 2 (1.9) | 1 (5.9) | |
| Days from onset to admission (in days) | 8.0 [7.0–10.0] | 7.0 [6.0–9.0] | 0.126 |
| Vital signs | | | |
| SBP, mmHg | 130.0 [118.0–139.0] | 139.0 [128.0–152.0] | 0.039 |
| HR, rate/min | 92.0 [81.0–101.0] | 90.0 [83.0–95.0] | 0.773 |
| BT,˚C | 37.6 [36.8–38.5] | 37.0 [36.8–38.1] | 0.351 |
| Respiratory condition on admission | | | |
| SpO$_2$, % | 94.0 [92.0–95.0] | 92.0 [88.0–94.0] | 0.022 |
| Respiratory rate, rate/min | 23.0 [20.0–28.0] | 23.0 [19.0–28.0] | 0.926 |
| Oxygen administered, n(%) | 103 (98.1) | 17 (100.0) | 1 |
| FiO$_2$ | 0.41 [0.33–0.50] | 0.50 [0.40–0.60] | 0.035 |
| SpO$_2$/FiO$_2$ ratio | 218.0 [186.0–285.0] | 184.0 [157.0–227.0] | 0.018 |
| ROX index | 9.9 [7.8–13.2] | 8.8 [6.9–9.9] | 0.071 |
| Pneumonia on initial chest CT, n(%) | 105 (100.0) | 17 (100.0) | NA |
| Therapeutic agents until admission day 3 | | | |
| Remdesivir, n(%) | 99 (94.3) | 17 (100.0) | 0.685 |
| Dexamethasone, n(%) | 105 (100.0) | 17 (100.0) | NA |
| Baricitinib, n(%) | 76 (72.4) | 11 (64.7) | 0.719 |
| Casirivimab/Imdevimab, n(%) | 6 (5.7) | 0 (0.0) | 0.685 |

Notes: Continuous variables are presented as medians with interquartile ranges in square brackets and were analyzed using the Mann–Whitney *U* test. Categorical variables are presented as numbers and percentages in parentheses and were analyzed using Fisher's exact test.

Abbreviations: HFNC, high flow nasal cannula; BMI, body mass index; COPD, chronic obstructive pulmonary disease; CHF, chronic heart failure; CKD, chronic kidney disease; SBP, systolic blood pressure; HR, heart rate; BT, body temperature; SpO$_2$, peripheral capillary oxygen saturation; FiO$_2$, fraction of inspired oxygen; ROX, respiratory rate-oxygen; CT, computed tomography.

NL ratio (6.70 vs 25.14, respectively, $p<0.001$), IL-6 (10.65 vs 30.3, respectively, $p = 0.003$), LDH (410.0 vs 636.0, respectively, $p<0.001$), and D-dimer (1.60 vs 7.00, respectively, $p = 0.041$) between the success and late failure group.

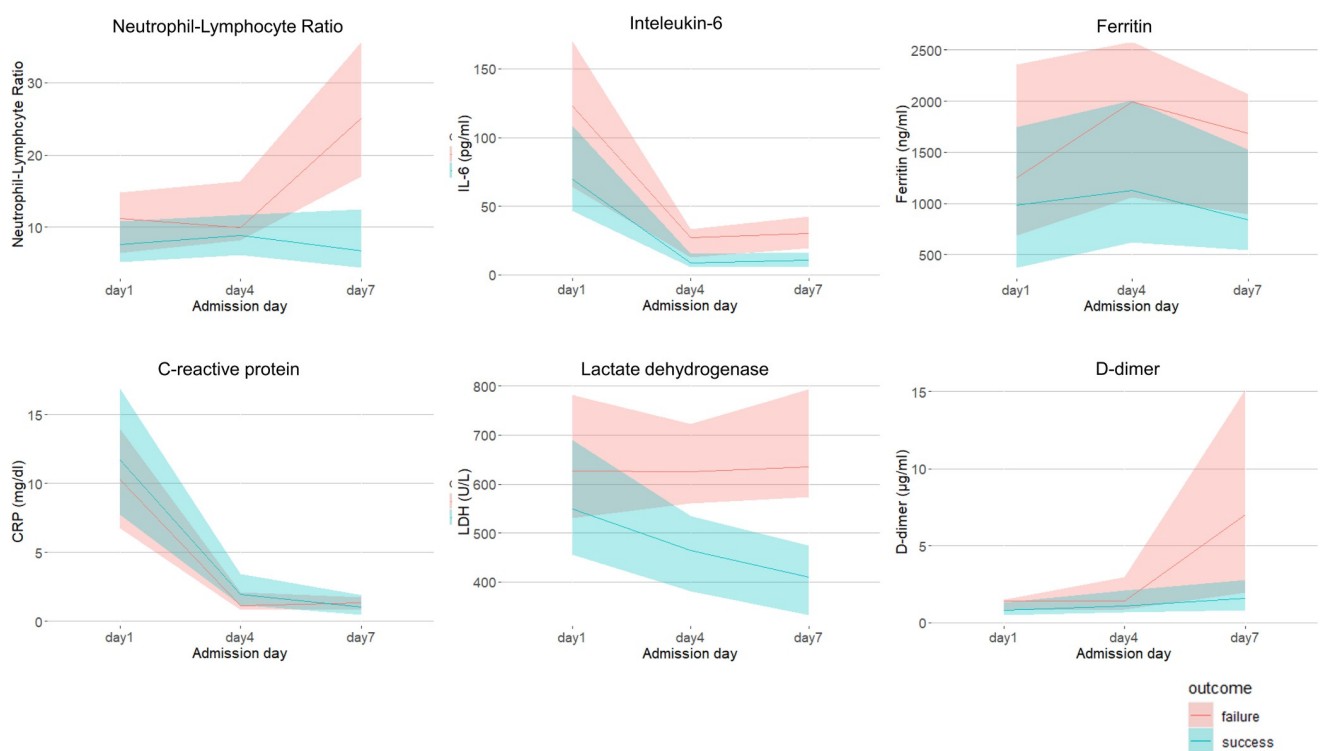

**Fig 2. Trends in each laboratory parameter and SARS-CoV2 RNAemia.** The line graphs show values of each laboratory marker stratified based on HFNC success (red) and failure (green) on Days 1, 4, and 7 presented as medians and interquartile ranges.

The univariate logistic regression analysis revealed a higher odds ratio for the late HFNC failure associated with LDH (OR: 1.42, 95%CI: [1.09–1.89]) and IL-6 (OR: 1.04, 95%CI: [1.00–1.19]) for each 10% change rate from Day 1 to Day 4. Conversely, no significant trends were observed for CRP (OR: 0.81, 95%CI: [0.54–1.05]), NL ratio (OR: 1.04, 95%CI: [0.97–1.10]), ferritin (OR: 0.99, 95%CI: [0.96–1.01]), and D-dimer (OR: 1.00, 95%CI: [0.97–1.01]) (Table 2). The AUROC for late failure on Day 4 indicated the highest LDH value at 0.82, followed by IL-6 at 0.78. None of the laboratory markers on Day 1 exhibited an AUROC of 0.7 or higher for overall failure (S1 Fig).

In a cohort of 107 patients who were tested for serum SARS-CoV2 RNA upon admission, the detection rate of serum SARS-CoV2 RNA was significantly higher in the failure group (13/

**Table 2. Univariate logistic regression analysis displaying unadjusted odds ratio for 10% change in lab markers on Days 1–4 of late high flow nasal cannula failure.**

|           | Unadjusted OR | 95% CI    |
|-----------|---------------|-----------|
| NL ratio  | 1.04          | 0.97–1.10 |
| CRP       | 0.81          | 0.54–1.05 |
| IL-6      | 1.04          | 1.00–1.19 |
| LDH       | 1.42          | 1.09–1.89 |
| Ferritin  | 0.99          | 0.96–1.01 |
| D-dimer   | 1.00          | 0.97–1.01 |

Abbreviations: OR, odds ratio; CI, confidence interval; NL, neutrophil-lymphocyte; CRP, C-reactive protein; IL-6, interleukin-6; LDH, lactate dehydrogenase

15, 86.7%) compared to the success group (37/97, 38.0%) (*p* = 0.001). Furthermore, the quantified log-transformed SARS-CoV2 RNA levels were markedly elevated in the failure group (2.37 [1.74–2.82]) compared to the success group (0.00 [0.00–1.54], *p*<0.001). Among the 108 patients who underwent serum SARS-CoV2 RNA testing on either Day 4 or 7, or both, 48 patients (44.4%) exhibited persistent SARS-CoV2 RNAemia. Notably, the late failure group demonstrated a substantially higher rate of persistent SARS-CoV2 RNAemia (11/11, 100.0%) compared to the success group (35/92, 38.1%) (*p*<0.001). Fig 3 illustrates the incidence rate of HFNC failure stratified by the presence or absence of SARS-CoV2 RNAemia. The survival analysis indicated a higher HFNC failure rate in patients with SARS-CoV2 RNAemia on admission (log-rank, p<0.001). Additionally, patients with persistent SARS-CoV2 RNAemia demonstrated a higher HFNC failure rate compared to those without persistent SARS-CoV2 RNAemia (log-rank, p<0.001). Even among the HFNC success group, a subset of patients with persistent SARS-CoV2 RNAemia (37/97, 38.1%) exhibited prolonged durations of HFNC therapy (7.0 [5.0, 9.0] days vs 5.0 [4.0, 6.3] days, respectively, *p* = 0.004) and extended hospital stays (17.0 [14.0, 26.0] days vs 13.0 [11.0, 15.0] days, respectively, *p*<0.001), in contrast to a subset of patients without persistent SARS-CoV2 RNAemia (60/97, 61.9%). In the correlation analysis between each laboratory parameter and SARS-CoV-2 RNA level, moderate correlation was noted specifically in LDH and RNA levels on both Day 4 (*r* = 0.549) and Day 7 (*r* = 0.468) (S2 Fig).

## Discussion

In an unprecedented environment, where the need for intubation was minimized through protocolized HFNC treatment for patients with moderate to severe COVID-19, the study cohort

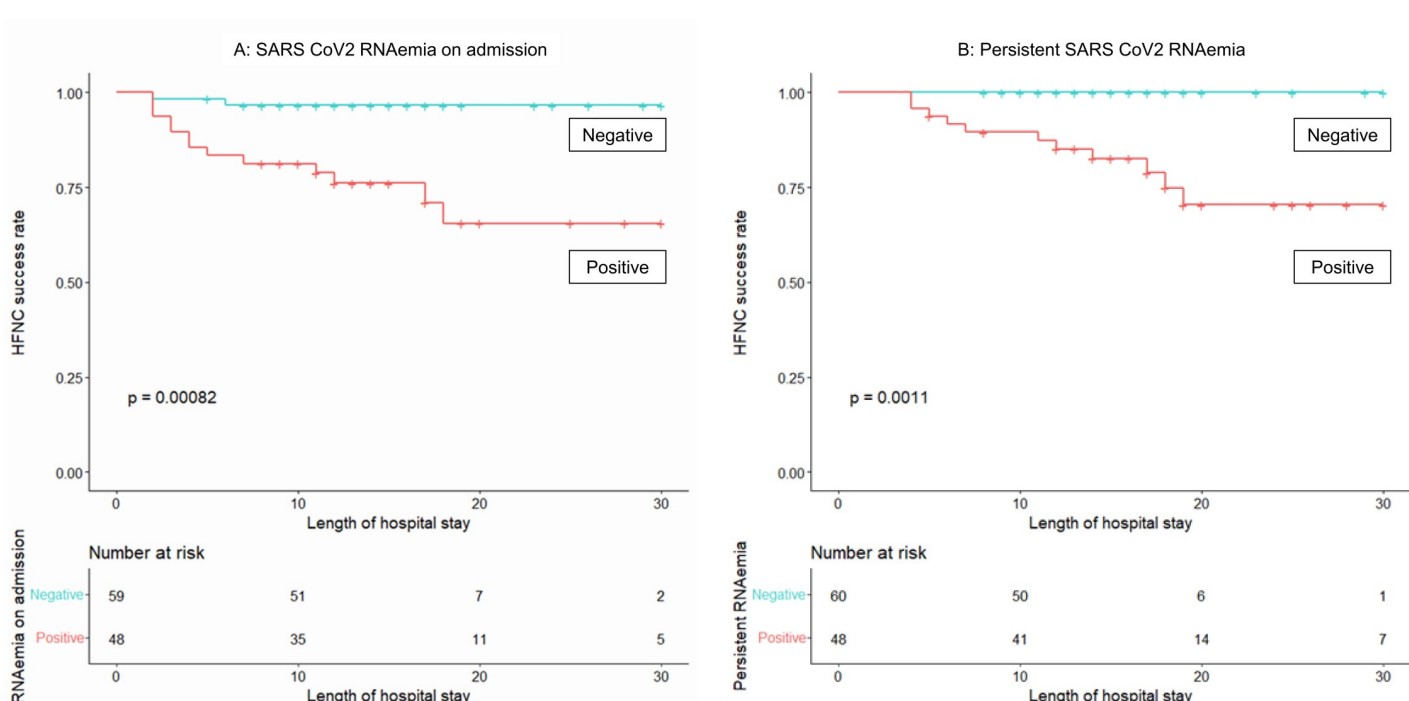

**Fig 3. Kaplan-Meier Curve depicting the incidence of HFNC failure, stratified based on the presence of SARS-CoV2 RNAemia.** A: Kaplan-Meier Curve showing the incidence of HFNC failure on admission in the overall study cohort, stratified based on the presence or absence of SARS-CoV2 RNAemia, along with the result of log-rank test. B: Kaplan-Meier Curve showing the incidence of HFNC failure in the subset of the study cohort excluding early failure, stratified based on the presence or absence of persistent SARS-CoV2 RNAemia, along with the result of log-rank test.

provides valuable insights into the effectiveness and therapeutic limits of HFNC, as well as the disease progression following treatment initiation. Of the various laboratory markers that predict the severity of the disease, chronological trends of LDH, IL-6, and SARS-CoV2 RNAemia, post-therapeutic intervention, were shown to be essential for predicting the success or failure of HFNC. These markers can also be instrumental in determining the appropriate timing for intubation.

The first point to note is that the levels of CRP and IL-6 exhibited a significant decline from Day 1 to Day 4 following induction of treatment, including anti-inflammatory drugs, irrespective of HFNC success. However, it was observed that IL-6 in the failure group displayed a comparatively lesser treatment response compared to the success group. Although cytokine storm through IL-6 is considered the key to disease progression [22], the higher levels in the HFNC failure group, even after treatment intervention in this study, suggest that relatively poor control of inflammation was a contributing factor to HFNC failure. Meanwhile, a notable difference in LDH trends was observed between the groups, with LDH levels in the success group consistently decreasing, while LDH levels in the failure group remained persistently high even after the treatment. Elevated LDH has been considered as evidence of hypoxic tissue metabolism and lung tissue damage, and correlated to the worse outcome [23–25]. Examining a cohort of 8,860 patients within the extensive Japanese COVID-19 registry highlighted a significant association between the LDH value on admission day 8 and in-hospital mortality [24]. A machine learning analysis using the same registry also showed that LDH level on admission strongly predicted HFNC failure rate compared to other factors [25]. In the present study, we clearly demonstrated that an upward trend in LDH before and after a treatment intervention, which included a pharmacological regimen not previously shown, could predict the success or failure of HFNC.

A notable strength of the study resides in our demonstration of quantified serial serum SARS-CoV2 RNA levels, facilitating a rigorous comparison of laboratory parameters and the treatment course. We found that both initial and persistent SARS-CoV2 RNAemia strongly correlated with HFNC failure. Furthermore, even in patients who successfully discontinued HFNC, the persistence of SAR-CoV2 RNAemia correlated with prolonged duration of HFNC treatment and extended hospital stays. Several studies showed that SARS-CoV2 RNAemia reflects the disease severity, by worsening host dysregulation and escalating lung tissue damage [16–18]. Interestingly, a previous study revealed that SARS-CoV2 RNAemia was correlated with high LDH and IL-6 values, in concordance with the findings of our study [15–17, 26]. These indicate progressive lung injury refractory to initial treatment, and are likely to require advanced respiratory support, suggesting that there is a justification for proceeding to invasive respiratory support, including IMV or extracorporeal membrane oxygenation (ECMO).

In our study cohort, the median $PaO_2/FiO_2$ ratio upon admission was 213 (mean 231), and the estimated $PaO_2/FiO_2$ ratio, calculated by substituting the HFNC induction criteria of SpO2 92% and oxygen flow rate of 4–6 L/min into the validated formula, was 185–284 [20]. Previous studies have shown that the failure rate of HFNC ranged from 43% to 70% when HFNC was initiated at a $PaO_2/FiO_2$ ratio of 105–224 [27, 28]. The HFNC failure rate in our study was 13.9%, which was remarkably lower than that in previous study cohorts. This may be attributed to the following reasons: potential inclusion of patients with milder disease severity or those admitted and administered drugs earlier following disease onset; and early initiation of HFNC may have also contributed to reducing the failure rate by alleviating the work of breathing and minimizing patient self-inflicted lung injury [29]. Meanwhile, HFNC was administered over an extended duration in the study cohort, raising the possibility that instances of intubation following prolonged HFNC treatment could have adversely influenced ultimate lung function and mortality [30]. Nonetheless, in this study, no patients experienced severe

complications either before or after intubation or during the transfer process, despite a higher incidence of complications being reported during intubation in COVID-19 cases [31].

The study has several limitations. First, this was a single-center retrospective cohort study conducted under specific circumstances, limiting its generalizability. However, the findings obtained in such a unique and irreproducible situation proved valuable in assessing the limitations of HFNC treatment. Second, the outcomes after patient transfer were not documented. Therefore, we could not evaluate the overall mortality and the accurate incidence of intubation. However, all patients with late failure except one were either intubated or died with respiratory failure, limiting the impact on the trend analysis. Third, the lack of documented trends in the ROX index at the time of or after HFNC initiation restricted direct comparisons with previous studies. Fourth, the data trends were measured based on time of admission and were not expressed in terms of days since the start of HFNC. This may have influenced the assessment of the relationship between data trends and HFNC success or failure. Finally, serum SARS-CoV2 RNA could not be measured in approximately 10% of patients due to insufficient specimen volume or order errors.

In conclusion, failure of HFNC treatment can be predicted in cases where there is a lesser reduction in LDH and IL-6 values or persistent SARS-CoV2 RNAemia after therapeutic intervention. Considering these factors in conjunction with physiological indicators may aid in establishing a more appropriate timing for intubation.

## Supporting information

**S1 Table. Baseline characteristics of the patients among three groups.**
(DOCX)

**S2 Table. Trends in each laboratory parameter and SARS-CoV2 RNAemia.**
(DOCX)

**S1 Fig. AUROC for HFNC failure in each laboratory marker.** The left-hand side of the graph depicts the AUROC for overall HFNC failure for each laboratory marker on Day 1, whereas the right-hand side shows the AUROC for late HFNC failure for each laboratory marker on Day 4.
(TIF)

**S2 Fig. Correlation between SARS-CoV2 RNA level and laboratory parameters.** The scatter plot and regression line with a 95% confidence interval (shown in gray) shows a correlation between log-transformed SARS-CoV2 RNA level (vertical axis) and laboratory parameters on Days 1, 4, and 7 (horizontal axis), with the Pearson correlation coefficient (r) indicating the strength and direction of the correlation.
(TIF)

## Author Contributions

**Conceptualization:** Hiroshi Koyama, Kazuya Sakai, Yoshihisa Aida, Tadashi Kamio, Takeru Abe, Mototsugu Nishii.

**Data curation:** Hiroshi Koyama, Kazuya Sakai, Kiyomitsu Fukaguchi, Mototsugu Nishii.

**Formal analysis:** Hiroshi Koyama, Kazuya Sakai, Kiyomitsu Fukaguchi.

**Funding acquisition:** Kazuya Sakai, Takeru Abe, Mototsugu Nishii, Ichiro Takeuchi.

**Investigation:** Hiroshi Koyama, Kazuya Sakai, Kiyomitsu Fukaguchi.

**Methodology:** Hiroshi Koyama, Kazuya Sakai, Kiyomitsu Fukaguchi, Takeru Abe.

**Project administration:** Hiroshi Koyama, Kazuya Sakai.

**Resources:** Hiroshi Koyama, Kazuya Sakai.

**Software:** Hiroshi Koyama, Kiyomitsu Fukaguchi.

**Supervision:** Takeru Abe, Mototsugu Nishii, Ichiro Takeuchi.

**Visualization:** Hiroshi Koyama, Kiyomitsu Fukaguchi.

**Writing – original draft:** Hiroshi Koyama.

**Writing – review & editing:** Hiroshi Koyama, Kazuya Sakai, Kiyomitsu Fukaguchi, Hiroki Hadano, Yoshihisa Aida, Tadashi Kamio, Takeru Abe, Mototsugu Nishii, Ichiro Takeuchi.

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
