## [Decision Letter · Decision Letter 0]

15 Mar 2024

PONE-D-24-01802A comparison study of temporal trends of SARS-CoV2 RNAemia and biomarkers to predict success and failure of high flow oxygen therapy among patients with moderate to severe COVID-19PLOS ONE

Dear Dr. Koyama,

Thank you for submitting your manuscript to PLOS ONE. After careful consideration, we feel that it has merit but does not fully meet PLOS ONE’s publication criteria as it currently stands. Therefore, we invite you to submit a revised version of the manuscript that addresses the points raised during the review process.

We look forward to receiving your revised manuscript.

Kind regards,

Ennio Polilli

Academic Editor

PLOS ONE

2. In this instance it seems there may be acceptable restrictions in place that prevent the public sharing of your minimal data. However, in line with our goal of ensuring long-term data availability to all interested researchers, PLOS’ Data Policy states that authors cannot be the sole named individuals responsible for ensuring data access (http://journals.plos.org/plosone/s/data-availability#loc-acceptable-data-sharing-methods).

Reviewers' comments:

Reviewer's Responses to Questions

**Comments to the Author**

1. Is the manuscript technically sound, and do the data support the conclusions?

Reviewer #1: Yes

Reviewer #2: Partly

2. Has the statistical analysis been performed appropriately and rigorously? 

Reviewer #1: Yes

Reviewer #2: Yes

3. Have the authors made all data underlying the findings in their manuscript fully available?

Reviewer #1: Yes

Reviewer #2: Yes

4. Is the manuscript presented in an intelligible fashion and written in standard English?

Reviewer #1: Yes

Reviewer #2: Yes

5. Review Comments to the Author

Reviewer #1: The article entitled “A comparison study of temporal trends of SARS-CoV2 RNAemia and biomarkers to predict success and failure of high flow oxygen therapy among patients with moderate to severe COVID-19” by Koyama et al. reports use of chronological trend of laboratory parameters including LDH, IL-6 and SARS-CoV2 RNAemia to predict HFNC failure and success. The manuscript is well-written and I feel that the manuscript is suitable for publication in PLOS ONE after the following issues are addressed:

1. For readers who are not familiar with SARS-CoV2 RNAemia, perhaps brief introduction on SARS-CoV2 RNAemia is needed.

2. IL-6, LDH and SARS-CoV2 RNAemia seem to differ between the failure and success group on or starting from Day 4 – authors need to state clearly that chronological trend of LDH, IL-6 and SARS-CoV2 RNAemia can be predictive markers of late HFNC failure but not early HFNC failure.

3. Table 1: Are the values median or average? The values in parenthesis – are these range or interquartile?

4. In Introduction, the authors stated that ROX index in the early hours of HFNC can be used to predict HFNC failure but the current paper lacks trends in the ROX index following the HFNC treatment. The authors have already addressed this as one of the limitations of study but without comparing the performance of the proposed markers (LDH, IL-6 etc.) to that of ROX index, it may be hard to convince the readers why other markers would be needed. Particularly considering that the ROX index, SpO2%, and SpO2/FiO2 ratio at the time of admission already exhibited significant differences between the failure and success groups (Table 1). Perhaps in addition to Kaplan-Meier Curve, ROC curve can be made and area under ROC curve can be calculated to show specificity and sensitivity of these markers.

Reviewer #2: This research study aims to investigate relationship between the failure of high-flow nasal cannula therapy in COVID-19 patients and blood parameters and level of SARs-CoV-2 RNA by performing the longitudinal study. The manuscript well organized and written clearly. The concern I have is on the results and data analyses detail.

Although the research appears sound, I have a few concerns that prevent me from accepting the manuscript at the present stage.

1. Please provide detail whether there is any guideline to separate early HFNC failure as <3 day and failure group (on or after day4)?

2. In page 19, please provide data explanation (Figure 2 and Table 2) on day 7 in the text.

3. Since it seems that the laboratory parameter and SARS-CoV2 RNAemia were different among early and late failure, please provide more detail characteristics of the patients between these 2 groups in Table 1 to understand whether there is any baseline difference that might affect patient outcome.

4. In S1 figure, there was a problem with the figure. Only part of figure was shown.

6. PLOS authors have the option to publish the peer review history of their article (what does this mean?). If published, this will include your full peer review and any attached files.

Reviewer #1: No

Reviewer #2: No

---

## [Author Response · Author response to Decision Letter 0]

9 May 2024

We wish to express our appreciation to the reviewers for his or her insightful comments, which have helped us significantly improve the paper.

Reviewer #1: 

The article entitled “A comparison study of temporal trends of SARS-CoV2 RNAemia and biomarkers to predict success and failure of high flow oxygen therapy among patients with moderate to severe COVID-19” by Koyama et al. reports use of chronological trend of laboratory parameters including LDH, IL-6 and SARS-CoV2 RNAemia to predict HFNC failure and success. The manuscript is well-written and I feel that the manuscript is suitable for publication in PLOS ONE after the following issues are addressed:

1. For readers who are not familiar with SARS-CoV2 RNAemia, perhaps brief introduction on SARS-CoV2 RNAemia is needed.

Reply: We supplemented the sentence with the clarification: “SARS-CoV2 RNAemia, which indicates the presence of SARS-CoV-2 viral RNA in the bloodstream.”

2. IL-6, LDH and SARS-CoV2 RNAemia seem to differ between the failure and success group on or starting from Day 4 – authors need to state clearly that chronological trend of LDH, IL-6 and SARS-CoV2 RNAemia can be predictive markers of late HFNC failure but not early HFNC failure.

Reply: In Table 2 and the result section of the univariate logistic regression, we emphasized “late” failure in the sentences.

3. Table 1: Are the values median or average? The values in parenthesis – are these range or interquartile? 

Reply: We added a footnote below Table 1 to provide clarification on the presented values as follows: “Notes: Continuous variables are presented as medians with interquartile ranges in square brackets and were analyzed using the Mann–Whitney U test. Categorical variables are presented as numbers and percentages in parentheses and were analyzed using Fisher’s exact test”.

4. In Introduction, the authors stated that ROX index in the early hours of HFNC can be used to predict HFNC failure but the current paper lacks trends in the ROX index following the HFNC treatment. The authors have already addressed this as one of the limitations of study but without comparing the performance of the proposed markers (LDH, IL-6 etc.) to that of ROX index, it may be hard to convince the readers why other markers would be needed. Particularly considering that the ROX index, SpO2%, and SpO2/FiO2 ratio at the time of admission already exhibited significant differences between the failure and success groups (Table 1). Perhaps in addition to Kaplan-Meier Curve, ROC curve can be made and area under ROC curve can be calculated to show specificity and sensitivity of these markers.

Reply: We acknowledge the reviewer's observation regarding the absence of a direct comparison between the trends of respiratory parameters and each laboratory marker, which could be considered a limitation of our study, as we also addressed in the discussion section. However, it's crucial to note that in real clinical settings, clinicians often base their judgments not solely on respiratory status but also on laboratory trends following pharmacological intervention. Hence, we believe that our findings offer valuable insights for clinicians. In response to the reviewer's suggestion, we conducted additional analyses to calculate the AUROC of each laboratory marker for predicting HFNC failure, as depicted in S1 Figure in the revised manuscript. The results revealed high AUROC values for LDH and IL-6 on Day 4 in predicting late failure, which aligns with the findings of the univariate logistic regression. These findings collectively suggest that monitoring the trends of LDH and IL-6 post-treatment initiation assists clinicians in estimating disease trajectory.

Reviewer #2: 

This research study aims to investigate relationship between the failure of high-flow nasal cannula therapy in COVID-19 patients and blood parameters and level of SARs-CoV-2 RNA by performing the longitudinal study. The manuscript well organized and written clearly. The concern I have is on the results and data analyses detail.

Although the research appears sound, I have a few concerns that prevent me from accepting the manuscript at the present stage.

1. Please provide detail whether there is any guideline to separate early HFNC failure as <3 day and failure group (on or after day4)?

Reply: There is no universal definition regarding the timing of HFNC failure to date. In the current study, we categorized the early failure group that failed before the 2-point blood tests on Day 1 and Day 4, and the late failure group that failed after the 2-point blood tests. 

2. In page 19, please provide data explanation (Figure 2 and Table 2) on day 7 in the text.

Reply: We added the data explanation on Day 7 in the result section as follows: “On Day7, following the exclusion of 5 patients who experienced HFNC failure between Day 4 and Day 7, a significant difference was observed in broader laboratory markers: NL ratio (6.70 vs 25.14, respectively, p<0.001), IL-6 (10.65 vs 30.3, respectively, p=0.003), LDH (410.0 vs 636.0, respectively, p<0.001), and D-dimer (1.60 vs 7.00, respectively, p=0.041) between the success and late failure group”.

3. Since it seems that the laboratory parameter and SARS-CoV2 RNAemia were different among early and late failure, please provide more detail characteristics of the patients between these 2 groups in Table 1 to understand whether there is any baseline difference that might affect patient outcome.

Reply: We further conducted a comparison among three groups: success, early failure group, and late failure group. In order to facilitate readers' understanding of the disparities in baseline characteristics, we separately documented these analyses in S1 table, in addition to Table 1.

4. In S1 figure, there was a problem with the figure. Only part of figure was shown.

Reply: We revised the legend of S2 figure (S1 figure on the initial manuscript) to provide clarification as follows: “The scatter plot and regression line with a 95% confidence interval (shown in gray) shows a correlation between log-transformed SARS-CoV2 RNA level (vertical axis) and laboratory parameters on Days 1, 4, and 7 (horizontal axis), with the Pearson correlation coefficient (r) indicating the strength and direction of the correlation”.

---

## [Editor Report · Decision Letter 1]

23 May 2024

A comparison study of temporal trends of SARS-CoV2 RNAemia and biomarkers to predict success and failure of high flow oxygen therapy among patients with moderate to severe COVID-19

PONE-D-24-01802R1

Dear Dr. Koyama,

We’re pleased to inform you that your manuscript has been judged scientifically suitable for publication and will be formally accepted for publication once it meets all outstanding technical requirements.

Kind regards,

Ennio Polilli

Academic Editor

PLOS ONE
---

## [Editor Report · Acceptance letter]

29 May 2024

PONE-D-24-01802R1 

PLOS ONE

Dear Dr. Koyama, 

I'm pleased to inform you that your manuscript has been deemed suitable for publication in PLOS ONE. Congratulations! Your manuscript is now being handed over to our production team.

Kind regards, 

on behalf of

Dr. Ennio Polilli 

Academic Editor

PLOS ONE